# Permafrost Hydrology Research Domain: Process-Based Adjustment

**Nikita Tananaev** [1,*] **, Roman Teisserenc** [2]  **and Matvey Debolskiy** [3] 

1   Melnikov Permafrost Institute, SB RAS, Yakutsk 677010, Russia
2   EcoLab, Université de Toulouse, CNRS, 31326 Toulouse, France; roman.teisserenc@toulouse-inp.fr
3   Geophysical Institute, University of Alaska Fairbanks, Fairbanks, AK 99775, USA; mvdebolskiy@alaska.edu
*   Correspondence: tananaevni@mpi.ysn.ru; Tel.: +7-4112-334-476

**Abstract:** Permafrost hydrology is an emerging discipline, attracting increasing attention as the Arctic region is undergoing rapid change. However, the research domain of this discipline had never been explicitly formulated. Both 'permafrost' and 'hydrology' yield differing meanings across languages and scientific domains; hence, 'permafrost hydrology' serves as an example of cognitive linguistic relativity. From this point of view, the English and Russian usages of this term are explained. The differing views of permafrost as either an ecosystem class or a geographical region, and hydrology as a discipline concerned with either landscapes or generic water bodies, maintain a language-specific touch of the research in this field. Responding to a current lack of a unified approach, we propose a universal process-based definition of permafrost hydrology, based on a specific process assemblage, specific to permafrost regions and including: (1) Unconfined groundwater surface dynamics related to the active layer development; (2) water migration in the soil matrix, driven by phase transitions in the freezing active layer; and (3) transient water storage in both surface and subsurface compartments, redistributing runoff on various time scales. This definition fills the gap in existing scientific vocabulary. Other definitions from the field are revisited and discussed. The future of permafrost hydrology research is discussed, where the most important results would emerge at the interface between permafrost hydrology, periglacial geomorphology, and geocryology.

**Keywords:** active layer; Arctic hydrology; cold regions hydrology; linguistic relativity; permafrost hydrology

## 1. Introduction

The Arctic is undergoing a sound transformation, affecting climate [1] and ocean temperature [2], sea ice extent [3], and terrestrial and marine biodiversity [4,5], driven by the Arctic amplification phenomenon [6–8]. Our understanding of long-term climate change effects on the Arctic water cycle is deduced from observations and reanalysis data [9], and relies heavily on existing forecasting and modeling experience, together with general assumptions elaborated mostly for temperate regions [10,11]. Where Arctic terrestrial hydrology is counted similar, coherent, or deducible by analogy with temperate regions, its regional uniqueness, or 'Arcticness', is questioned. However, the Arctic is a frontier ecosystem with distinct features, where permafrost and related effects are known to play an important role.

The hydrological boundaries of the Arctic region are loosely defined by the basin approach [12]. The Arctic hydrology domain is, hence, extended southward up to the smallest headwater streams of the northern Mongolia, as a pan-Arctic drainage area [13,14]. Integrated across millions of square kilometers of drainage basins, the output signals of hydrological processes and human impacts are transmitted to the Arctic ocean margin, feeding input to a complex marine system, and impacting global

oceanic freshwater turnover and chemistry [15–17]. This integrated approach, coined by marine science, blurs the physiographic diversity of the pan-Arctic basin, and a potential diversity of hydrological response throughout the region.

Linking Arctic amplification to Arctic hydrology is complicated by an ambiguity of 'the Arctic' definition, put into hydrological context. One should acknowledge that the Arctic amplification and its effects on terrestrial hydrology occur in a very particular region in the high latitudes, regardless of its exact limits. They cannot be understood by analogy with temperate regions, primarily because of the presence of permafrost. The latter has an enormous effect on the water cycle, as most hydrological processes are confined to nonfrozen layers in an otherwise frozen media [18–20]. The direct linkage between the thermal state of permafrost and the heat and water fluxes is a unique regional feature [21–23]. Only in permafrost regions, water phase transitions resulting from long-term changes in temperature and/or precipitation definitively affect the hydraulic properties of soils [24].

Permafrost hydrology, as a distinct research field, from its very beginning aimed at better understanding and quantifying these interconnections between frozen ground and hydrological processes [25,26]. In a changing Arctic, attention is growing toward the role hydrology plays in the organic matter and nutrient transport [27] and permafrost–climate feedback [28]. Long-term upward trends in the active layer thickness can potentially liberate up to ca. 800 Pg of highly degradable organic carbon stored in perennially frozen soils [29,30], the fate and transport of which depend on the hydrological processes in the active layer and in Arctic streams [31,32].

However, the attempts to summarize the current state of knowledge in permafrost hydrology are relatively scarce [11,19,33,34]. A recent review paper by Walvoord and Kurylyk [35] provides a comprehensive overview of the major terms and fundamental concepts of permafrost hydrology, mostly related to permafrost (active layer, talik, etc.). Further advances in permafrost hydrology may require researchers to align their understanding of the discipline domain, research objectives, and methods. This brief paper discusses the limits of relevance of permafrost hydrology, as a branch of modern geophysics, in the field of regional and global change research. We attempt to redefine the permafrost hydrology domain through a process-based adjustment, and introduce several concepts relevant to future studies in the field.

## 2. The Definition of Permafrost Hydrology and Linguistic Relativity

Language is frequently argued to act as an active cognitive tool, both in sciences of humanity and in physics [36]. Science is operating concepts, theories, and models, which are all language-based cognitive abstractions. As the late A. Einstein [37] put it, " . . . the mental development of an individual and his way of forming concepts depend to a high degree upon language". Our mode of thought, comprehension, and cognition is bound to our native language and its structure, the statement known as the Sapir–Whorf hypothesis [38]. A non-universal nature of the current knowledge originates in part from the international character of science, the development of certain research fields in different language environments. Research in Earth sciences, including both geocryology and hydrology domains, took differing directions in Soviet and Occidental scientific traditions [39]. Subsequently, in the scientific language of these schools, either the literally identical term is understood differently, or different untranslatable terms do exist. Both cases require a "vocabulary alignment", establishing a common framework for informal discussions and collaboration. In permafrost research, the basis for such vocabulary alignment was established in 1988 in Canada with a glossary for English permafrost-related terms [40]. Later, it was translated to multiple languages, including Russian, and officially recognized by the International Permafrost Association in 2005 [41]. Nonetheless, even if the vocabulary agreement was achieved grosso modo by this publication, there still exists substantial divergence in current usage of various permafrost-related terms. In the case of 'permafrost hydrology', both words in the phrase are known to be understood and used differently.

*2.1. Permafrost*

Permafrost is a layer of soil, rock, sediment, or other earth material with a temperature that has remained at or below 0 °C for two or more consecutive years, irrespective of its lithology or water/ice content [41,42]. This definition remains unchanged, virtually unchallenged, and is widely used in Occidental literature [43]. Put simply, it allows the researchers to conclude on the presence at a given point (e.g., pit, borehole) and, by extension, at a given depth, of a cryotic (at or below 0 °C) material that represents, and embodies, permafrost. This definition assigns a term to a specific thermal state of soil or rock parcel rather to an object or event class per se [44], and by extension, to a certain geological stratum, a three-dimensional volume of frozen material. It yields certain material and temporal aspects, but otherwise is largely void of context, and has to be placed in such to be properly understood. As such, it makes part of several constructions, denoting:

(1) Generic properties: Permafrost + (thermal state); the only example since permafrost is defined uniquely through temperature;

(2) Physical properties: Permafrost + (temperature, ice content); otherwise, 'frozen soil' is used in references to particular material properties, such as heat transmissivity, hydraulic conductivity, unfrozen water content, etc.;

(3) Spatial aspect: Permafrost + (region, area, zone, extent, distribution), also 'permafrost type'; (subsea/submarine, mountain, lowland, continuous, isolated) + permafrost, addressing physiographical settings, and continuity;

(4) Geological features: Permafrost + (base, table, thickness, texture);

(5) Temporal evolution: Permafrost + (genesis/origin, dynamics, development, degradation/aggradation, thawing), though never 'permafrost freezing'; and

(6) Relative or possessive case: Permafrost + (construction, foundation, map, model, soil, carbon pool, loss).

In Russia, the definition of 'permafrost' was at the heart of a significant discussion among the researchers in the 1940s and early 1950s. The competing opinions were that permafrost be defined (1) through its thermal state, i.e., as cryotic rock regardless of ice content, or (2) through the presence of ice in the pore space. The latter point was judged more scientifically solid, as phase transitions and the presence of ice in soils and rocks ultimately change their thermal, geoelectrical, and geomechanical properties and permeability. The generally accepted definition of permafrost is 'rocks and soils, having negative or zero temperature, in which at least some part of water is in crystalline form' [45]. Russian scientific language discerns different aspects of permafrost, regarding it:

(1) As a phenomenon, or rock or soil state: *merzlota* (mʲɪrzlɐtˈa), perennially frozen ground, or rather *mnogoletne-merzlye porody* (mntextschwaɡɐlʲˈetnʲɪ-mʲˈθrzlɨjtextschwa pɐrˈodɨ), perennially frozen rock/soil;

(2) As a territory underlain by those, fully or partially: *kryolitozona*, (krˈiolʲitɐzˈontextschwa), cryolithozone or permafrost zone;

(3) A three-dimensional geological body: *mnogoletne-merzlaya tolscha* (mntextschwaɡɐlʲˈetnʲɪ-mʲˈθrzlɐjtextschwa tˈolɕɕɐ), perennially frozen rock layer).

Here 'kryolitozona' connotes a specific physiographic region, whereas 'mnogoletne–merzlaya tolscha' emphasizes the vertical dimension of a frozen layer, though the two definitions are often used interchangeably in the Russian literature [45]. Another specifically Russian term that is currently in use worldwide is 'kryosfera' (krˈiosfʲˈɛrtextschwa), the cryosphere, encompassing parts of the Earth's crust, hydrosphere, and atmosphere, subject to temperatures below zero for at least part of each year [41]. This term was proposed in early 1920s by Polish researcher A.B. Dobrowolski [46] and then re-introduced to scientific language by P.A. Shumsky [47]. The cryosphere is defined by the presence of ice, and as such includes permafrost, but also glaciers and ice sheets, lake and river ice etc., appearing intimately overlapping with hydrosphere.

The main point where the two definitions really differ is in their regard to cryopegs, ground layers, or, more frequently, closed ground volumes or lenses, remaining nonfrozen at negative temperatures

because of the abundance of dissolved solids in the pore water [41]. Cryopegs may or may not be considered as parts of permafrost, depending on how the latter is defined. They are mostly closed systems without direct contact with groundwater, and, thus, are not directly connected to a hydrological network.

## 2.2. Hydrology

The comparable level of ambiguity exists in defining 'hydrology' across languages. In the English usage, 'hydrology' refers mainly to a research discipline preoccupied with water, but can also be used to reference the totality of water-related processes, and/or water budget of a particular water body. The most frequent use includes constructions referring to:

(1) Methods and applications: (Statistical, isotope, engineering, computational, contaminant) + hydrology, these are all separate research fields or disciplines;

(2) Water cycle elements: (Surface water, groundwater) + hydrology;

(3) Compartments: (Surface, subsurface, soil, active layer) + hydrology;

(4) Landscapes and specific objects: (Prairie, forest, peatland, floodplain, glacier, subglacial, periglacial) + hydrology, implies the specific water cycling processes in these ecosystems; though extremely rarely 'tundra hydrology', and never 'lake/river hydrology';

(5) Spatial aspect: (Land, catchment, watershed, drainage basin) + hydrology;

(6) Frozen water: (Snow, ice, meltwater) + hydrology; though never 'rain hydrology';

(7) Particular water bodies: Hydrology of the + (Pacific Ocean, Lena River, Great Lakes; each water body can have its proper hydrology;

(8) Particular regions: Hydrology of the + (Everglades, Polk County, Northern Carolina, Arctic).

In Russian scientific vocabulary, this term has coherent meaning, but distinctly different usage. The prevalent use is oriented towards generic objects void of spatial context, hence 'river/lake/reservoir/wetland hydrology' are all legitimate terms for disciplines studying these water bodies per se. Landscapes, as connoting landcover classes, are never covered by the term, thus 'forest hydrology' is absent from Russian vocabulary. Methods and applications form disciplines' names in virtually the same style, e.g., 'engineering hydrology'. Likewise, distinct water objects can have its own hydrology, e.g., 'the Danube Delta hydrology', but not the regions, thus 'Arctic hydrology' is an illegitimate term. However, 'Arctic Ocean drainage basin hydrology' may serve as a close substitute, coherent to the modern vision [11], and underscoring the fact that there is much less of the Arctic in our 'Arctic hydrology' speech than we would normally assume.

## 2.3. Permafrost Hydrology

In Occidental literature, permafrost hydrology is a research discipline, studying " . . . the direct and indirect effects of perennially frozen ground on the properties, occurrence, distribution, movement and storage of water" [25]. This definition is an elegant attempt to express what comes evident from our reasoning, that permafrost hydrology is studying hydrology in permafrost. As such, it mimics the constructions like 'forest hydrology', and maintains a notion to permafrost as a specific landcover (or 'undercover') or ecosystem class. Therefore, no Russian analogue could exist (see above). Since permafrost is cryotic rock/soil, the term interestingly aligns with the use of 'hydrology' with other cryotic substrates, snow and ice. Hydrological processes in permafrost environments are thought to be " . . . not unique to [permafrost] districts, but their intensities differ from those in temperate latitudes" [25].

In Russian usage, as it was discussed above, no hydrology exists for ecosystem classes, and this discipline name took form 'permafrost rivers hydrology', or *gidrologiya rek kryolitozony* (gɪdrɐlˈ ogɪjtextschwa rʲˈek krˈiolʲitɐzˈonɨ), in a late 1980s Russian text by B.L. Sokolov, later published in English as 'Hydrology of rivers of the cryolithic zone in the U.S.S.R.' [48]. Eventually, '*gidrologiya kriolitozony*' (permafrost hydrology) was used, though infrequently, in Russian literature,

and a permafrost hydrology laboratory was active at the State Hydrological Institute (Saint Petersburg) from the early 1990s until 2007.

Perennially and seasonally thawed layers, such as taliks and notably the active layer, are excluded from permafrost by definition. However, it is frequently stated that 'most biogeochemical and hydrological processes in permafrost are confined to the active layer' [49]. The active, or seasonally thawed, layer is inherent to permafrost, as are the taliks, neither making part of it, however. These are all layers void of permafrost by definition. Whatever is occurring in these layers, occurs outside permafrost. Hydrological processes are nevertheless only active in these nonfrozen media; that said, do they really take place in permafrost? The preceding discussion clarifies this issue. Permafrost can be regarded not only as a frozen or cryotic ground, but as either an ecosystem class or a region, *kriolitozona*, in which case the hydrology of this specific ecosystem is addressed.

An interesting conclusion emerging from this linguistic analysis is that neither English nor Russian languages are efficient in defining and explaining permafrost hydrology. In English vocabulary, 'hydrology' can widely denote a totality of terrain-specific processes inherent to water cycle, but the 'permafrost' definition as cryotic rock/soil is too narrow. In Russian, vice versa, 'permafrost' has long been viewed as a separate region, but 'hydrology' has too restricted usage to be applied to such regions. Therefore, a successful definition of permafrost hydrology relies on coupling these two approaches, treating permafrost as an ecosystem class and hydrology as an ensemble of water-related processes, typical for such ecosystems.

How can we define whether a specific region or process has anything to do with permafrost hydrology? Hydrological objects vary greatly in size, and the largest Arctic drainage basins can have their headwaters either in discontinuous permafrost regions, or nonpermafrost areas. Which share should permafrost hydrology claim in these cases, and how may we discern it from other hydrologies?

## 3. Permafrost Hydrology: Process-Based Definition

In a permafrost domain, water finds its way from precipitation, snowmelt, or ground ice meltdown to streams, subjected to the universal action of gravity, much as elsewhere on Earth. Viewed as a region, permafrost (*kriolitozona*) hosts processes, unrelated to the frozen ground influence on water transfer and storage. Universal physical laws are governing water movement at any given point, but a specific set, or 'assemblage', of hydrological processes may exist, defining which laws are the most applicable, and which forces are dominating, under particular conditions and in particular landscapes. This approach is not uncommon to Earth sciences. Geomorphology acknowledges specific landforms as imprints of a particular (set of) geomorphic processes or events, e.g., palaeo-ice stream beds [50], and defines the environment through landform assemblages, notably the definition of paraglacial environment [51].

Permafrost can be perceived as a specific ecosystem class, thence, arguably, there exists a typical assemblage of hydrological processes, through which a research domain of permafrost hydrology can be defined. The unique process assemblage for permafrost hydrology includes: (1) Water table migration caused by upper aquitard development through freeze–thaw processes; (2) water migration in soil matrix, driven by phase transitions in the active layer; (3) transient water storage in solid state in the subsurface compartment. Implied processes are not solely hydrological, and may also be regarded as cryogenic, related to phase transitions in soils [52], or periglacial, driving landscape conditioning by frost action [53].

### 3.1. Water Table Migration

Thawing front, or the active layer base, acts as a cryogenic aquitard for the percolating water [54]. Infiltration is restricted by frozen soil or, under certain conditions, in frost heave-susceptible soils, by segregation ice lenses formed during freezing [55]. Hydraulic conductivity of frozen soils depends on the volume of pores occupied by ice: It varies from 15% to 35% of the nonfrozen soil when ice content is between 0.3 and 0.5 p.u., and declines to almost zero at ice content over 0.6 p.u. [56]. Rapid decline

in hydraulic conductivity is in part related to selective freezing of soil pores, when larger pores freeze earlier owing to lower capillary forces [57].

The thawing front position within the soil column evolves through a temporal continuum of states, as shown in Figure 1, and hydrological processes follow this evolution [18,35,58]. The annual cycle starts in late winter when the active layer is frozen and the subsurface compartment is virtually void of moving water. Solar radiation penetrates the snow cover [59], and early thawing in topsoil can proceed under snow. Subsequently, this thawed layer partly accommodates meltwater runoff during spring.

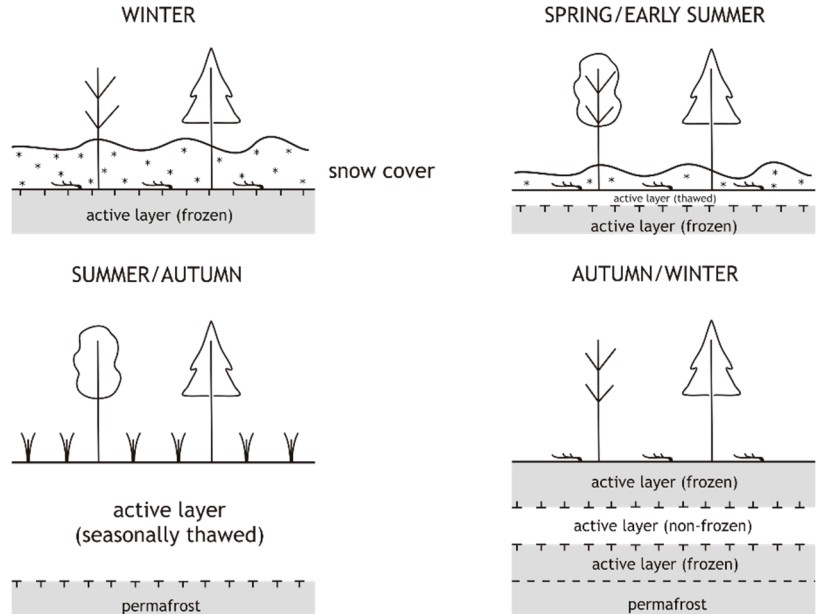

**Figure 1.** Active layer is a dynamic feature, whose state is constantly evolving. Snapshots of typical and illustrative states are shown, though intermediate states do exist and form a temporal continuum. Frozen ground is shaded grey, and dashed line with directed ticks shows its limits; ticks are directed toward frozen ground. Dashed line denotes the top of permafrost.

Seasonal thaw develops throughout the warm period, and groundwater table variations during this time can be caused solely by the thawing front propagation. At the moment of maximum annual thawing, late summer or early autumn, the only aquitard corresponds to the active layer base which coincides with the top of permafrost (Figure 1). Later in autumn, when soil starts refreezing, the active layer base disengages from the top of permafrost. The active layer de facto accommodates two aquitards, (1) frozen top-soil surface, (2) active layer base, which is also a local freezing front. The former limits late autumn rainwater infiltration, while the latter accumulates available water to form an ice-rich transient layer [60]. The unfrozen soil stores water throughout the freezing period, and may reroute it toward the streams in late winter or early spring [61].

Permafrost virtually eliminates loss to infiltration and cuts off deep ground water recharge, except by talik zones. Catchment response to storms, therefore, depends less on antecedent conditions, and active layer depth is an important factor. Active layer development leads to decreasing water table slope and hydraulic head in the subsurface compartment, affecting stream–groundwater interaction.

### 3.2. Soil Water Migration

Water movement in the nonfrozen parts of the active layer mostly occurs under gravity force action, as elsewhere. Besides, the active layer freezing provokes both vertical and lateral water migration in nonfrozen soils under the action of several freezing-related processes.

Vertical water migration to the freezing front is observed in fine-grained soils and peat [62,63], while in coarser material, water is forced to migrate downward from the freezing front [64]. Pore

pressure excess in the residual active layer during late autumn and early winter (Figure 1) promotes lateral migration of water in the saturated layer, what is called 'piston flow' in Russian literature [65].

Ice lenses developed in soil through ice segregation serve as local freezing fronts, hence several regions of upward and/or downward migration can be present in the active layer at any given moment. Consequently, multiple local desiccation zones are developed in the soil profile, causing differential compaction and cracking, and resulting in an overall increase in vertical permeability [65] and hydraulic conductivity [66].

*3.3. Transient Water Storage*

Permafrost is capable of redistributing water fluxes, acting in a wide range of timescales. Seasonally, water can be captured in the active layer as textural and segregation ice in winter, to be released in spring and summer upon active layer thawing. In clastic material, e.g., blockfields or kurums, spring meltwater freezes up in large pores and is released as summer advances [67]. Icings are typical permafrost hydrology features that can capture, store, and redistribute groundwater on the timescale from seasons to several years or even decades [48,68,69].

The runoff volume intercepted by icings can be as high as 12% to 22% of total basin discharge under continuous permafrost conditions [70]. Water trapped in an ice-rich transient layer will only be released upon continuous climate warming, which is on timescales from hundreds to thousands of years [60]. Ice wedges and textural or massive ice, e.g., in the Ice Complex deposits or buried glaciers, can be preserved throughout millennia before re-entering the global water cycle when exposed in river banks or marine cliffs or subject to thermokarst degradation [71,72].

Whether 'permafrost hydrology' is viewed as a specific research discipline or refers to the water cycling through a particular ecosystem, this process assemblage defines the limits of its research domain. Other processes may be involved in cycling water in these environments, but we believe these three processes maintain the particularity of permafrost hydrology. To our viewpoint, if any of these processes is missing, either from the researchers' considerations or from the natural landscape, this places the research outside the permafrost hydrology domain.

From this position, we consider regions with seasonally frozen ground to be out of the scope of permafrost hydrology, since the active layer base serves as an aquitard for only short period of a year, and after its disappearance, the water percolates into deeper horizons and is lost to water cycle, as elsewhere on Earth. The same line holds for regions with fossil permafrost, i.e., Western Siberia, where the top of the permafrost is too deep to interfere with water transport, except associated groundwaters. In contrary, the areas with two-layered permafrost where the top of permafrost is close to the active layer base, i.e., Northern Yenisey region or Subarctic European Russia, especially where the separating talik is occasionally refreezing in some years, are considered within permafrost hydrology domain.

## 4. Permafrost Hydrology: Spatial Domain

Hydrological studies are generally based on basin approach. An elementary watershed is the smallest response unit, and water-routing processes are spread and averaged across. Though some processes are studied at a stand scale, e.g., infiltration or transpiration, observed river runoff is attributed to a certain catchment area, where water routing and transfer do occur.

Numerous studies concerning permafrost hydrology and defining its current state were conducted in discontinuous and even sporadic permafrost regions, wherever permafrost underlies the full areal extent of the research site [73]. An effective budget-saving strategy, this approach requires a certain degree of coalescence between the permafrost extent and the study area extent, to be relevant as a permafrost hydrology study.

To formally define if a study can be distinguished as a permafrost hydrology study, we suggest a strategy described further in Table 1. The applicability of this rule depends on the relative scale of the study compared to the extent of the global permafrost regions defined by continuity criteria [74]. For the small-scale studies, at least the active layer and top of the permafrost should be included in

the domain, so that all the processes associated with the active layer could be accounted for. The mesoscale (ca. 25–2500 km$^2$) catchments are included in the permafrost hydrology domain when their surface is underlain by continuous permafrost. In discontinuous permafrost, the existing taliks under lakes and channels are to be closed or, if these are open taliks, they should play a significant confining role for the subpermafrost aquifer [75]. On the global basin scale, the presence of permafrost-affected subcatchments should be acknowledged, and relevant cryogenic processes should be taken into account.

**Table 1.** Permafrost hydrology applicability in relation to both permafrost and study extent.

| Study Area Extent | Area | Permafrost Extent | | | |
|---|---|---|---|---|---|
| | | Patchy <10% | Sporadic 10–50% | Discontinuous 51–90% | Continuous >90% |
| Stand plot | point | Yes | Yes | Yes | Yes |
| Slope; representative elementary watershed (REW) [76] | <10 km$^2$ | No | Yes | Yes | Yes |
| Mesoscale watershed; hydrological response unit (HRU) [77] | <2500 km$^2$ | No | No | Yes | Yes |
| Macroscale watershed and global basins | >2500 km$^2$ | Permafrost-affected HRUs should be explicitly described or modelled as such | | | |

## 5. Hydrologies in the North: Definitions, Existing and Revised

Permafrost hydrology adds to other disciplines already present in the field; their domains may occasionally overlap, creating confusion and misunderstanding. To this end, this section overviews existing definitions of numerous hydrologies, and proposes their revision.

### 5.1. Existing Definitions

Research fields emerging from acknowledgement of a highly regional character of hydrological processes include: Arctic hydrology, cryohydrology, cold-regions hydrology, high-latitude hydrology, northern hydrology, periglacial hydrology, permafrost hydrology, and polar hydrology. These terms are at times used interchangeably, without giving a notice, and the difference is largely unclear and confusing.

'Arctic hydrology' is widely used to denote the processes in the Arctic Ocean and atmosphere, where terrestrial subsystem plays a minor role [10,14]. Recently, 'Arctic terrestrial hydrology' was thoroughly reviewed, its terrestrial contributing area expanded well beyond Arctic river basins, while Arctic freshwater domain embraces the good part of the Northern Hemisphere, including atmospheric and marine compartments [10,11]. This definition leaves aside the hydrological processes specific to the Arctic regions, defining Arctic freshwater domain on a catchment basis, i.e., including Northern Mongolia and washing out specific effects of permafrost on the water cycle.

The 40° N latitude roughly serves as a southern limit for the 'cold-regions hydrology', though other specific criteria are imposed on the definition of a cold region [19]. In the same logic, yet another term, 'cryohydrology', was recently coined by M.-K. Woo [78], explained as 'hydrology at low temperatures' and 'hydrology of cold regions'. 'High-latitude hydrology', in its turn, envisions its southern limit at 60° N latitude [52], sharing it with 'northern hydrology' in its earlier definition [79]. Lately, 'northern hydrology' has been encompassing the processes in the tundra and taiga ecosystems [80], or postglacial settings in boreal, temperate coniferous, and mixed forests [81]. None of these disciplines deals specifically with permafrost or cryogenic processes.

'Periglacial hydrology' was used to address the processes in the periglacial sector of the Vatnajökull ice cap [82]. 'Periglacial' has long been denoting areas bordering existing glaciers, but has now its sense expanded to areas adjacent to past glaciers, or simply to cold regions [53]. As such, it was used

to term something close to 'cryohydrology' in a paper by Johansson et al. [83]. Since permafrost is in the heart of periglacial landscapes, 'permafrost hydrology' may stand for periglacial hydrology in the hydrologist's eyes. Nevertheless, the term 'periglacial' is most frequently related to geomorphology, and has in its turn a long story of scientific debate behind (see, e.g., [53] and multiple references therein). For these reasons, we prefer 'permafrost' to 'periglacial', when speaking about hydrology.

'Polar hydrology' was a recent research effort of Norwegian Water Resources and Energy Directorate in Svalbard [84], for which no strict definition was given.

This brief review of existing definitions illustrates the dominant use of geographical borders, either randomly assigned or based on ecosystem limits, to discern between various definitions. To our opinion, from the hydrological standpoint, this approach is not viable, as the involved water-related processes do not change with, or because of, latitude change. This ambiguity is challenging for both communicating the research outcomes to the wider audience, and designing the balanced and focused research plans/projects. This motivation drives our efforts in revising these definitions.

### 5.2. Revised Definitions

The Arctic region is an important part of the global system, and as such receives influence from many distant regions, which in their turn should not be included in the Arctic domain just on this purpose of influence. Reiterating the discussion above, to us the 'Arcticness' is an important regional feature not to be neglected. 'Acknowledge, not include' should be a proper strategy in defining the Arctic domain: The influence of multiple distant non-Arctic regions, e.g., the headwaters of major Arctic rivers, and processes, e.g., El Niño/Southern Oscillation (ENSO) or Pacific Decadal Oscillation (PDO) [85], should be properly accounted for in research design and media discussions, but they should not be regarded as parts of the Arctic domain. Northern Mongolia is not the Arctic, even though it makes part of a pan-Arctic drainage basin.

"The Arctic is delineated by latitude, tree lines, national and subnational borders and indigenous territories, among many other suggestions" [86] page 1. However, only the national boundaries are clearly demarcated, and the view of the Arctic as a political region is arguably more correct in the scope of water research. Put simply, there are too many hydrologies defined on physiographical basis, thanks to a dominating 'basin approach', that we clearly need one based solely on national and subnational borders.

In this case, 'Arctic hydrology' should focus on freshwater resources in the Arctic, including studies of water quantity, availability, and quality, or, generally speaking, of water resources of the Arctic and its administrative regions, thus better serving the people of the North. Flowing water knows no national borders, but water resource management and planning are based on political borders, as controlled by Arctic states. Our approach aims at supporting Arctic states in adopting water management strategies for their northernmost regions even when the majority of the state surface is situated outside the Arctic, like, e.g., in Russia or the U.S.

The Arctic water resources assessment and planning are important in the scope of the Arctic change. The development and implementation of Arctic water resources vulnerability index serves an example of a regional research effort aiming to better understand and evaluate the risks to communities related to Arctic change [87].

Ecosystem or ecoregion boundaries are used to define northern, or boreal, hydrology as concerned with tundra and taiga ecosystems, and interactions between vegetation communities and water fluxes [80]. 'Northern' requires a supplementary definition of 'southern hydrology', be it either the Antarctic or Mediterranean climate, and is discouraged from use. Hence, 'boreal hydrology' sounds more appropriate, as connoting the existence of typical boreal ecosystems, such as forests (taiga) and wetlands.

'High-latitude hydrology', as a definition, is subject to a question of which latitude is high enough, in the hydrological sense. High latitudes are felt 'high' thanks to a rough climate. Therefore, we propose to define high-latitude hydrology as related only to the harshest Köppen E climate regions, including

any regional study concerning particular rivers or watersheds, and emphasizing particular site-specific features, such as highly seasonal flow, snow redistribution impact, etc. As such, it makes part of a larger 'cold-regions hydrology' domain, where cold regions are also defined based on climate [14,19].

With progressive climate change, warming the Arctic at an impressive rate, these definitions will also change their spatial coverage. The boundaries of Köppen climate types and ecosystem limits will shift northward, and regions previously considered as cold are expected to become warmer [88]. High-latitude and cold-region hydrology could be, therefore, considered as endangered species. We should track vigilantly the hydrological effects of these changes; this is, to our opinion, one of the most important objectives of permafrost hydrology research.

## 6. Future Progress in Permafrost Hydrology Domain

Permafrost hydrology is already a well-established research discipline as well as a general framework for the scientific advancement and planning, even though the use of the term is undermined by a strong blend with other terminologies. Upon defining its scope and objectives, its major concern should turn to the development of discipline-specific methods, best suited for particular research purposes. These methods and approaches, in our opinion, should address the permafrost-specific hydrological processes (see Section 3).

The active layer is hosting the majority of hydrological processes in permafrost regions, and is, as such, an important future research focus. From this perspective, the vertical water migration during the active layer freezing is by far the most promising research area. As previously discussed in the literature, soil moisture can move either to or from the freezing front, depending on the soil lithology [62–64], but the physical basis of both processes is poorly understood. Rapid progression of the freezing front down the soil profile leads to development of thin segregation ice lenses which impede infiltration during the next snowmelt season and could promote surface and fast subsurface runoff at that time.

Runoff distribution between different soil compartments is important in understanding the catchment reaction to rain events. In this scope, various tracer hydrology methods are promising in acquiring information on flow paths and residence times in permafrost catchments [80,89]. Freeze–thaw processes leave a distinct imprint on $^2$H and $^{18}$O signatures in soil moisture that can be traced to specific locations (by soil type), processes (e.g., ice segregation), ground ice forms (pingo, aufeis, ice wedges), or, speculatively, even to certain cryostructures. Rare earth elements are used to track hydrologic pathways, distinguishing fast and slow subsurface flows, or mineral or organic layers [90]. These data should be used in combination with field data on active layer dynamics, water table, vegetation, and precipitation, including rain chemistry, in assessing the activity of certain flowpaths and its spatio-temporal variability.

Organic carbon and its transformation affected by hydrological processes is a subject of utter importance in the scope of permafrost–climate feedback. There are multiple direct links between permafrost hydrology and biogeochemistry. Methane production is an anaerobic process, depending on the permafrost thaw rate and the water table position in the soil profile ([91], and references therein). Old permafrost-derived carbon is highly biodegradable, and is rapidly consumed by bacteria in a headwater system [92,93]. Optical and molecular properties of dissolved organic carbon can be used as quasi-conservative natural tracers to track the reaction of slopes and headwaters to progressive active layer development. Changes in active layer thickness owing to global change or local disturbance may result in substantial alteration of stream hydrology and ecology because of switching to more diverse feeding sources, increased sediment and nutrient fluxes, and channel pattern shifts [94,95].

Hydrological modelling strategies accounting for permafrost-specific processes should be developed or further enhanced for their better representation. The most promising approaches include: (1) Coupled water and heat balance models of various dimension and complexity, e.g., the 'zero-dimensional' model of Boike et al. [22], PFLOTRAN [96], SUTRA 3.0 (aka SUTRA-ice) [97] and permaFOAM [98], to mention but a few; (2) an explicit two-dimensional heat transfer model with phase

transitions, where water and heat fluxes are decoupled [99]; and (3) semi-distributed models with a simplistic permafrost description as an impermeable layer, and no heat transfer module [100,101].

Permafrost is present in Arctic river channels and banks, and river flow interaction with this frozen substrate has yet received little attention in the literature. Frozen bank stability and retreat rates were recently studied both in Alaska and central Yakutia [102–104], but no adequate model exists to date to describe or forecast thermal erosion of Arctic river banks. This subject is important since increased thermal erosion rates are expected in the warmer future, threatening riverside communities along the alluvial rivers in Arctic Russia and North America. Sub-riverine permafrost and its effects on channel pattern development is a 'blank spot' in modern permafrost hydrology research [105].

Water movement and redistribution in the subsoil compartment, along with phase transitions and related volumetric changes, necessarily perform geomorphic work. This complex overlap gives birth to a certain cryo-fluvial interaction [106], where periglacial landforms develop as a function of local hydrology, while surface runoff reuses and reshapes linear forms created by cryogenic processes. Ultimately, integrated environmental models should be developed describing the water routing in the subsoil, its heat imprint and geomorphic consequences, the development of non-channelized drainage network (water tracks) and thermo-erosion gullies, and its environmental effects [107,108]. Mass movement driven by cryogenic processes within the valley bottom limits may serve an important sediment source for permafrost rivers [95,109], dominating their total sediment flux under certain circumstances.

Mountains may host permafrost patches outside cold climate regions, hosting a variety of periglacial processes [110]. Even in the absence of glaciers, mountain permafrost may significantly influence the hydrological response in this highly dynamic environment, but has still received limited attention. Icing hydrology in relation to permafrost should particularly benefit from novel observation techniques, e.g., unmanned aerial surveys and radar satellite imagery.

## 7. Conclusions

Hydrological processes in periglacial environments are often regarded as " ... azonal processes operating in cold environments", and as such are believed to " ... differ little, if at all, from similar processes in other climatic environments" [53]. They are thought to be " ... not unique to [permafrost] districts, but their intensities differ from those in temperate latitudes" [25]. From what is known, the conclusion is that permafrost hydrology can be defined through a process assemblage, unique for permafrost regions. This assemblage includes: (1) water table migration controlled by active layer thawing, (2) water migration in the soil matrix driven by freezing front propagation, and (3) transient water storage in catchments in the form of ice. These processes make 'permafrost hydrology' different from other research disciplines. Other hydrologies do exist, and, having their definitions and scopes revised, could provide extremely important insights. This paper offers such revision, and other revisions may follow, to the benefit for conscious science, and the Northern communities.

Scientific advancement is constrained by the absence of common language and untranslatability of major 'local' terms coined by different national schools. This constraint can only be overruled by acknowledging the existence of multiple meanings in national scientific domains, and cross-linking these meanings, by discussion and consensus. Advancement of research and collaboration is sustained by this terminological framework, elaborated by the researchers in order to transfer and share their expertise. Clear research objectives and hypotheses arguably serve science not less than our sophisticated field and modeling methods.

**Author Contributions:** All authors contributed to the text. Conceptualization and writing, original draft preparation, N.T.; writing, review and editing, N.T., R.T., and M.D. All authors have read and agreed to the published version of the manuscript.

**Funding:** This research was partially funded by the Russian Fund for Basic Research, project No. 17-05-00948a (N.T.), TOMCAR-Permafrost Marie Curie International Reintegration Grant FP7-PEOPLE-2010-RG, project reference: 277059 (R.T.).

**Acknowledgments:** The paper was conceived and partially written while N.T. served as an invited professor to EcoLab, Université de Toulouse, France. The support and assistance from EcoLab staff are highly appreciated.

**Conflicts of Interest:** The authors declare no conflict of interest. The funders had no role in the design of the study; in the collection, analyses, or interpretation of data; in the writing of the manuscript, or in the decision to publish the results.

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
