# Peer review of "Permafrost Hydrology Research Domain: Process-Based Adjustment"

_hydrology, doi:10.3390/hydrology7010006_

Round 1

Reviewer 1 Report

1. line 108: Geological features should be supplemented with a

cryolithological description: structures, textures, genesis of

underground ice.

2. At Fig.1 the "Autumn/winter" part contain the inproper sign of the

active layer (AL): in fact, this is unfrozen part of the AL.

3. Table 1 is non informative and may be deleted without prejudice to

the content of the article. The article does not provide data

substantiating the contents of this table.

Author Response

The authors appreciate your valuable comments to the manuscript, which were addressed in the revised manuscript.

Lines 108 & 109 were supplemented with additional terms, like permafrost texture as geological feature, and permafrost origin/genesis as a feature of temporal evolution. Fig 1 was amended following your suggestions. Indeed, during autumn and winter, the frozen and non-frozen zones are still parts of the active layer. Though it was suggested to remove Table 1, we stand however to retain it in the main text, since it implicitly relates several important hydrological concepts, such as 'representative elementary watershed' and 'hydrological responce unit', to permafrost continuity.

Reviewer 2 Report

the authors attempt to redefine the term "permafrost hydrology", but I get the impression that they have difficulty with understanding both permafrost and hydrology. In addition, additional studies require how to define a specific concept. Therefore, the article is not insightful and based rather on superficial knowledge. I would suggest deeper studies before attempting to redefine terms again. In its current form, it does not contribute much, although you can notice interesting threads. I do not recommend publishing it without major corrections. More details in the text

Author Response

The authors are extremely grateful to the Reviewer who raised multiple issues crucial to the manuscript. The manuscript text was corrected accordingly, except points where, most probably owing to linguistic relativity, the Reviewer's comments were slightly out of scope of the general discussion.

We the authors are a permafrost hydrologist, a biogeochemist in aquatic environments, and a permafrost researcher. Our knowledge is mostly superficial, as in our research we mostly deal with surface waters and shallow permafrost layers close to the Earth surface. This will not necessarily mean we have no clear idea of what 'permafrost' is, or what 'hydrology' is. The problem is that neither the definitions are perfect, nor the understanding of these definitions is perfect as well. This is one of the main reasons this manuscript was conceived and written.

As the Reviewer's report have shown, there are substantial difficulties in defining and understanding terms in our field of science. Some of the Reviewer's comments serve in itself excellent proofs for the linguistic relativity concept applied to permafrost hydrology, and geography/geophysics in general. The knowledge, indeed, is language-dependent, and we are not the first to state this, and not the last. Large number of publications exist on the topic, supporting our point of language-dependent knowledge.

In the last lines, we reiterate our gratitude to the Reviewer for raising important questions and being overall attentive to our reasoning.

Please see the attachment for a complete discussion on the matter, including line-by-line response to Reviewer's comments.

Reviewer 3 Report

I have read the article submitted for review with great interest. This is not a text presenting new research or measurement results, but rather a discussion on how to understand and combine concepts in the field of hydrosphere research in areas where permafrost occurs. The text gives the opportunity to track the terminological and semantic differences of the term "permafrost hydrology", depending on the research traditions of various scientific communities. For English-speaking readers, it is also an opportunity to get acquainted with the terminology used by Russian researchers, which is sometimes not spread outside of publications in Russian. The authors attempted to confront the concepts of hydrology and permafrost research in Russia and Anglo-Saxon countries.
The narrative leads to several important problems, one of which is not such an unequivocal definition of permafrost. For me, the most interesting of the topics discussed concerns permafrost in the sea coast zone. However, it is not widely developed. If the authors see the possibility of expending their text, I would see here a brief reference to the problem of areas where the cryotic state of the ground with the limited presence of ground ice is found - cryopegs. I have no other objections to the scientific argument. I believe that the article could be enriched with at least one, two block diagrams or flow charts that would show the problems taken or the path of the argument led by the authors. Unfortunately, I am not able to assess its language correctness. Best regards.

Author Response

The authors greatly appreciate the interest in our manuscript, expressed by the Reviewer. The manuscript was extended to specifically mention the problem of cryopegs, noted by the Reviewer.

This manuscript is a resubmission of an earlier submission. The following is a list of the peer review reports and author responses from that submission.